# When Institutional Plates Collide: The Dynamic Impact of Informal Institutions on Capital Market Development

Robert Lindorfer [1], Anne d'Arcy [2,*] and Igor Filatotchev [3,4]

1    Austrian Federal Ministry of Finance, Johannesgasse 5, 1010 Vienna, Austria
2    Department of Strategy and Innovation, Vienna University of Economics and Business, Welthandelsplatz 1/D5, 1020 Vienna, Austria
3    King's Business School, King's College London Bush House, 30 Aldwych, London WC2B 4BG, UK
4    Department of Global Business and Trade, Vienna University of Economics and Business, Welthandelslatz 1/D1, 1020 Vienna, Austria
*    Correspondence: anne.darcy@wu.ac.at

**Abstract:** We provide an institutional theory perspective to examine societal legitimacy in the context of capital market development. While prior research has focused on the importance of formal institutions, firms are embedded within broader socio-economic structures associated with informal institutions. Using content analysis and a unique dataset of 3244 newspaper articles between 2004 and 2013, we develop a dynamic measure capturing the public perception of capital markets as a proxy of informal institutions. We run a Prais–Winsten regression with panel-corrected standard errors to explore the dynamic relationship between public perception of capital markets and equity market size in Austria and Poland. We further theoretically and empirically explore how formal and informal institutions mutually reinforce each other in the context of capital market development. Our results suggest that informal institutions matter differently in developed and emerging economies.

**Keywords:** capital market development; formal and informal institutions; societal legitimacy; institutional theory; developed and emerging economies

## 1. Introduction

In this study, we theoretically and empirically explore the role of informal institutional factors in the context of capital market development using content analysis and a unique dataset capturing public perception of capital markets as a dynamic and novel proxy for informal institutions. We aim to find evidence for why, despite its maturity and relatively advanced technological infrastructure, the Austrian capital market has gradually lost its leading position in Central and Eastern Europe (CEE) to the emerging Polish capital market. Austria, as an icon of continental Europe's corporatist tradition (Gourevitch and Shinn 2007), and Poland, as a transition economy aiming towards a liberal version of capitalism in the aftermath of the European Union (EU) enlargement in 2004 (Dobija and Klimczak 2010), seem an excellent laboratory for studying the impact of public perception of capital markets as an informal institutional factor affecting capital market development.

Prior financial economics literature has identified formal institutions associated with formally codified rules and regulations, such as investor protection, and macroeconomic factors such as income, and trade openness to be the most important forces supporting capital market development (Billmeier and Massa 2009; La Porta et al. 1997, 1998; Rajan and Zingales 2003; Smaoui et al. 2017). However, much attention has been paid to formal institutional factors such as national and international regulations (La Porta et al. 1997, 1998), whereas informal institutional factors such as normative and cognitive legitimacy among wider groups and population constituencies have hardly entered the academic discussion. By showing that, and under which circumstances, institutional reforms are favorable for national capital markets, our main contribution is to add public perception

of capital markets as an important endeavor to explain capital market development in a particular economy.

We make an important methodological contribution by measuring informal institutions beyond cultural dimensions. While many researchers have presumed that informal institutions are constant, prior work has suggested that this assumption may not be valid (Cantwell et al. 2010; Meyer and Peng 2005; Sartor and Beamish 2014). By applying content analysis to the measurement of informal institutions, we use a dynamic measure capturing normative and cultural-cognitive legitimacy. The measurement further implicates a different understanding of the resilience of informal institutions: we argue that informal institutions may be subject to continuous change.

This study makes contributions to theory in the fields of international finance and economics. First, we add an informal institutional factor to the explanation of capital market development. As capital market development provides an aggregated view of the extent to which market actors actually use capital markets, we further link informal institutions to firms' financing strategies. Although the influence of informal institutions on organizational behavior is not completely new in the field of international finance (Hartwell and Malinowska 2019; Holmes et al. 2013; Kwok and Tadesse 2006), we aim to explain the impact of a country's formal and informal institutional factors on its capital market development. Second, studies linking informal institutions to strategic organizational responses often associate informal institutions with culture (Kwok and Tadesse 2006; Salomon and Wu 2012). Although culture may be an important determinant of the level and evolution of informal institutions, shared cognitive structures evolve through prior cognitive processes and should be considered within a dynamic institutional perspective (Meyer and Peng 2005; Wrona et al. 2013). As public perception of capital markets is difficult to operationalize, we apply content analysis to the measurement of informal institutions and use a novel and time-variant measure capturing normative and cognitive legitimacy (Gaur and Kumar 2018). This dynamic institutional perspective offers a more complete measurement of national differences and environmental complexity than a rather static cultural perspective (Cantwell et al. 2010; Chang 2011). Third, considering informal institutions in addition to formal institutions adds to the discourse on the complex interrelationships between institutional factors (Cruz-García and Peiró-Palomino 2019; Holmes et al. 2013; Lewellyn and Bao 2014). We aim to resolve the disparate findings and refine existing theory through adding a novel measure of informal institutions. Fourth, comparing developed and transition economies explains the influence of the initial level of economic development on the relationship between informal institutions and macro-economic outcomes (Aixalá and Fabro 2008; Lee and Kim 2009; Smaoui et al. 2017). Because an increased level of uncertainty is particularly present in a transitional context, the extent to which informal institutions influence capital market development is expected to differ between Austria and Poland.

Our comparative empirical analysis builds on quarterly time-series data from the Austrian and Polish equity markets between 2004 and 2013. This period is especially appropriate for the study of equity market development, as financial globalization became particularly salient (Doidge et al. 2013), and data availability and comparability in Poland were improved with the EU accession. We chose equity market size to measure capital market development that we interpret as a field-level proxy for firms' financing strategies. We use content analysis applied to 3244 newspaper articles, published in the leading business press of Austria and Poland, to measure public perception of capital markets as a proxy for informal institutions, and the Heritage Foundation's Index of Economic Freedom as a proxy for formal institutions. To explore the dynamic relationship between public perception of capital markets and equity market size in Austria and Poland, we run a Prais–Winsten regression with panel-corrected standard errors. Although we are convinced that our research setting already eliminates several endogeneity concerns, we choose additional macroeconomic control variables to strengthen the validity of the data analysis.

In contrast to previous research that has focused primarily on individual differences in formal institutions, we explore how the entire set of political, legal, and economic policies,

in addition to the informal dimension of public perception of capital markets, work as the determinants of equity market size. We hypothesize and find evidence that, indeed, informal institutions affect equity market size beyond the influence of formal institutions. In that respect, our results add academic support that public perception of capital markets plays an important role in the development of capital markets through motivating and constraining firms' financing strategies. By measuring the interaction effect between formal and informal institutions, our results further confirm that the greater the economic freedom, the higher the positive relationship between public perception of capital markets and equity market size. Moreover, our results report that the influence of informal institutions is more important for Austria, where the low level of the public perception of capital markets hinders capital market development.

The study is structured as follows. First, we elaborate on the empirical setting and develop an institutional perspective on capital market development by integrating relevant literature and theory to develop our hypotheses. Next, we present the data and research methodology, followed by the empirical results. Finally, we discuss the study results and present its implications and limitations.

## 2. Empirical Setting

In the field of informal institutions and international finance, transition economies of CEE provide a highly suitable source of information, as they had to build up their capital markets in the context of changing public perceptions after the fall of the communist regimes. Specifically, since the EU enlargement in 2004, the Polish capital market has gradually gained legitimacy among investors through harmonized regulative and transparency efforts (Grosfeld and Hashi 2007). When we conducted a series of in-depth, semi-structured interviews with capital market insiders in Austria and Poland, a top executive of a leading Austrian bank explained:

> "... it [Polish capital market development] started building up very fast from the moment Poland entered the European Union, which was the breaking point for Polish companies to leverage their business potential."

Within this period, actors in the Austrian capital market also expected additional growth opportunities through the economic potential of CEE, as many firms in transition economies were desperately seeking financial resources to support their restructuring and modernization. The European integration has increased the number of regional capital markets that national and international firms considered when making their listing decisions. Despite their similar strategic intents, both capital markets differed in their formal institutional and macroeconomic development. In contrast to Austria, historically known as a rather mature and regulated capital market, Poland has implemented several profound structural and regulatory changes since the EU accession. Dobija and Klimczak (2010) provide an overview of the Polish transition focusing on the implementation of International Financial Reporting Standards (IFRS) and its code of corporate governance. They describe the development of the Warsaw Stock Exchange starting in 1991 to become a medium-sized stock exchange with a leading position in Central and Eastern Europe. In contrast, the Austrian capital market has failed to become an economic gate to Eastern Europe. Despite its maturity and relatively advanced technological infrastructure, the Austrian capital market has gradually lost its leading position in CEE to the emerging Polish capital market, especially in the aftermath of the 2008 financial crisis (Figure 1 illustrates the different development during the period under investigation and especially during the financial crisis). This clearly indicates that the presence of technological infrastructure and high levels of financial savings per capita may be a necessary but not sufficient condition for the development of a country's capital market.

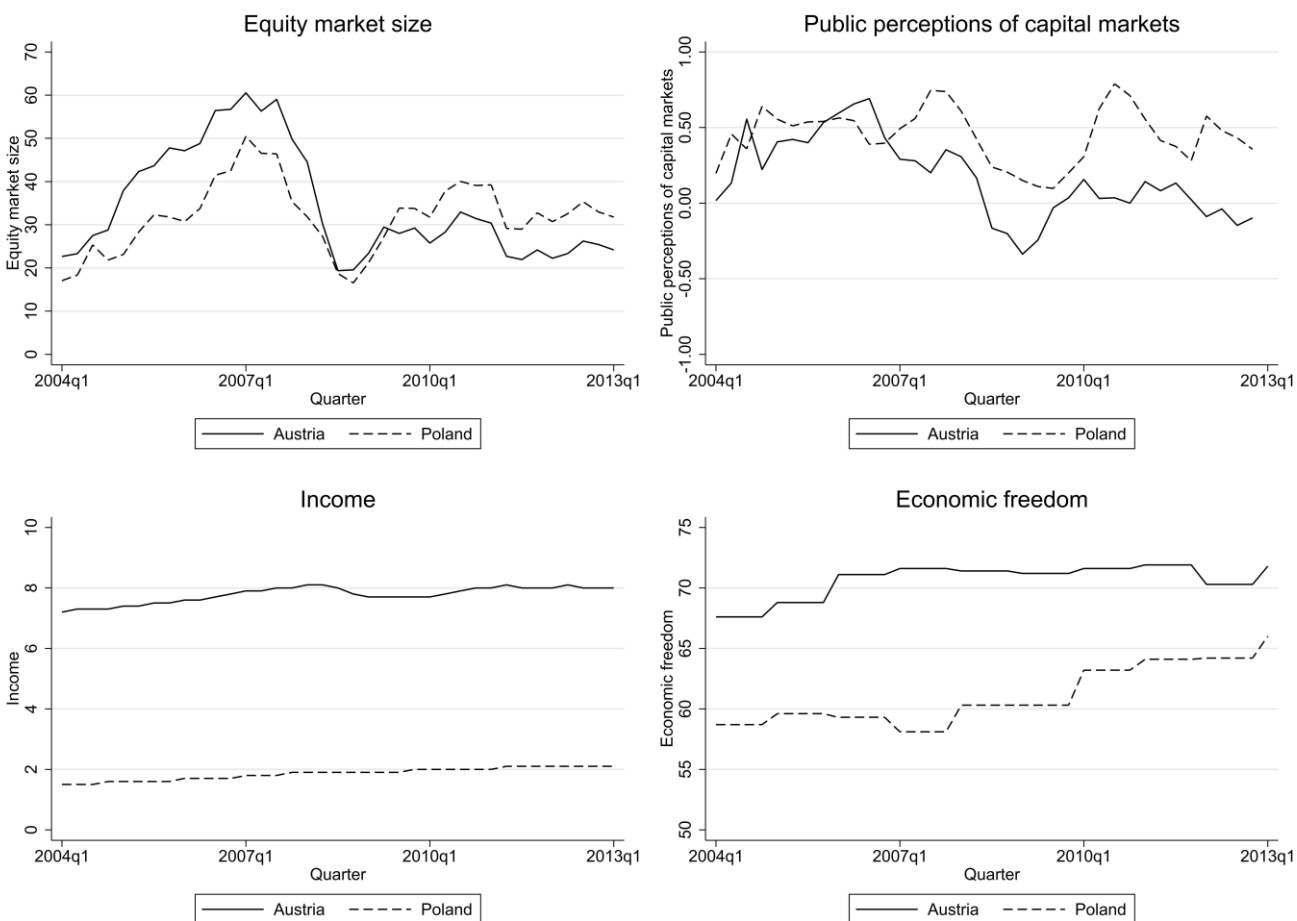

**Figure 1.** Comparative time series graphs of main regression variables. Data sources: Equity market size (equity market capitalization as a share of real GDP per year) and Income (real GDP per capita in constant EUR thousandth) are taken from the Eurostat database, economic freedom is measured by the Heritage Foundation's Index of Economic Freedom, and Public perception of capital markets is measured through Janis–Fadner coefficients of coded newspaper articles from the Factiva database.

It is crucial to consider the differences in institutional settings between Austria and Poland. Austria, as a representative of the coordinated, insider-oriented, and relationship-based market economies of continental Europe, has not developed institutionalized norms and beliefs similar to those in the liberal, outsider-, and market-oriented Anglo-American model of capitalism (Meyer and Höllerer 2016). As an example, the Austrian Stock Corporations Act contains an obligation to stakeholder orientation in corporate governance that requires the executive board of a corporation to act, above all, in the best interest of the corporation, thereby considering the interests of shareholders and employees as well as the public good. Poland, by contrast, already accords with a model of liberal capitalist economies, and it relies heavily on the use of markets for economic coordination (Redek and Sušjan 2005; Dobija and Klimczak 2010). Another interview revealed that Polish society has a more positive perception of capital markets:

> "[... the Warsaw stock exchange] has support from the whole country, which is actually behind the growth of the capital market."

Considering this specific institutional environment, we aim to provide first empirical evidence of the impact of societal legitimacy on capital market development.

## 3. Literature Review, Theory and Hypotheses Development

### 3.1. Literature Review and Theoretical Framework

We apply an institutional theory perspective to examine legitimacy in the context of capital markets. Legitimacy is "a generalized perception or assumption that the actions of an entity are desirable, proper, or appropriate within some socially constructed system of norms, values, beliefs, and definitions" (Suchman 1995, p. 574) and is therefore a form of social judgment, existing only in the minds of external assessors (Bitektine 2011). Accordingly, legitimation is the process of making a practice or institution socially, culturally, and politically acceptable within a particular context (Suchman 1995). Following institutional theorists, we distinguish between formal and informal institutions: formal institutions refer to laws and regulations of a particular country, and informal institutions are supported by values, beliefs, customs, traditions, and codes of conduct (North 1990; Salomon and Wu 2012). Formal institutions are intended to provide rules for economic transactions and to regulate the behavior of the actors involved in economic exchange (North 1990). Because a robust legal environment protects potential financiers against expropriation by entrepreneurs, it raises their willingness to provide funds in exchange for securities, and hence expands the scope of capital markets (La Porta et al. 1997). Legal differences in investor protection across countries shape the ability of insiders to expropriate outsiders and thus determine investor confidence in markets and consequently market development (La Porta et al. 1997, 1998; Shleifer and Wolfenzon 2002). Hence, differences in legal protection help explain why firms are financed and owned so differently in different countries (La Porta et al. 1998). However, firms are embedded within broader social structures, comprising different types of institutions that exert significant influence on firm strategies (Holmes et al. 2013; Peng et al. 2009). Besides government actors and regulators, research has highlighted the role of the public in endorsing and conferring legitimacy (Guiso et al. 2004; Deephouse 1996, 2000). National societies consist of shared cognitive structures that shape how people view the world, determine how they make sense of events occurring in that world, and help them interpret the explanations offered by others (Chui et al. 2002; Zilber 2006). Informal institutions are patterns of common behavior and socially shared rules and elicit shared cognitive and normative frameworks among economic actors (Helmke and Levitsky 2004). The classification into formal and informal institutions further offers three advantages for our analysis. First, the different institutional factors are clearly defined and avoid any confusing overlap (Arslan and Larimo 2010). Second, the neo-institutional framework focuses on national institutions and thus lends itself to a country-level analysis (Kostova and Roth 2002). Third, this approach allows for consistency and clarity in analyzing differences within national contexts (Estrin et al. 2009).

A growing body of research in international finance and economics is exploring factors that affect capital market development and consequently firms' financing strategies in individual countries, as developed factor markets in general, and capital markets in particular, are important building blocks of modern economic societies (Cumming et al. 2017; Moore et al. 2012; Zajac and Westphal 2004). Financial development is a central determinant in explaining various economic and societal phenomena, such as the occurrence of banking crisis and economic growth (Naceur et al. 2019; Levine and Zervos 1998). At present, capital market development is further increasingly used for measuring institutional quality in explaining firms' internationalization strategies (Driffield et al. 2016; Hoskisson et al. 2013). We chose equity market size to measure capital market development as using equity market size provides the advantage that the extent to which market actors actually use capital markets is directly reflected (Demirgüç-Kunt and Levine 1996).

Prior research has identified that macroeconomic factors and legal rules and their enforcement matter for the size and extent of a country's capital market (Djankov et al. 2008). Although these studies emphasize the impact of institutions on capital market development, they predominantly focus on the role of formal institutions (Li 2013; La Porta et al. 1997, 1998). Informal institutions have only sparsely begun to enter the research field, mostly in the context of product markets (Hartwell and Malinowska 2019; Krause et al.



2016; Sartor and Beamish 2014). Therefore, there is a lack of studies on how a complex interface between formal and informal institutions affects national factor markets, including capital markets. This opens up a research opportunity to develop a more holistic theoretical approach linking institutions and factor market development. In this study, we aim to address these theoretical and empirical gaps.

We suggest an institutional theory-grounded research framework to understand the societal legitimacy mechanisms behind firms' listing decisions in different institutional contexts. A central premise is that the type of uncertainty precipitated by informal institutional differences is a significant determinant of strategic firm behavior. The few existing scholarly attempts have treated institutions as static and have asked what responses social actors would enact as a result of a relatively stable institutional configuration (Chang 2011; Li 2013). Instead, we consider institutionalization as a dynamic process and examine how capital market development responds to changing institutional rules and social beliefs in specific countries. Furthermore, we explore the complex interplay between formal and informal institutions in the context of a developed and a transition economy and their effect on the behavior of capital market actors. This is an important research setting, as transition economies have relatively less-developed formal institutions, and, as a result, economic actors may rely on their perceptions of informal rules and norms when making strategic decisions (Park 2021). Many transition economies are moving away from state control and build some key institutions of modern capitalism from scratch, including functioning capital markets.

### 3.2. Informal Institutions and Capital Market Development

Traditionally, scholars refer to informal institutions, broadly, as cultural institutions (Lewellyn and Bao 2014; Salomon and Wu 2012), where culture is often defined as a system of shared values, beliefs, and attitudes that influences individual perceptions and behaviors. Although culture may be an important determinant of the level and evolution of informal institutions, shared cognitive structures evolve through former cognitive processes such as perception, expectation formation, development of different alternatives in strategic choice, or learning activities (Wrona et al. 2013). Hence, we question the static concept of culture often used to measure informal institutions, as cognitive processes and external circumstances continually change the cognitive structures of national societies (Hoffman 1999). The main reason to expect institutions' contribution to capital market development is that they provide external coordination mechanisms, entail productivity-enhancing incentives and decrease transaction costs through reduced uncertainty for economic actors (Hartwell 2013; Kingston and Miguez 2009; Williamson 2000). In addition, a more positive public perception of capital markets is expected to make the listing decision a more legitimate course of action. Acting in conformity with public perceptions conveys an accepted framework that legitimates firms' financing strategies, reducing individual blame in the case of failure. A more positive public perception of capital markets further increases the likelihood that other firms will behave similarly (Kostova and Roth 2002), again making the listing decision a socially accepted course of action. This corresponds with the assumption that equity market size is positively correlated with the ability to mobilize capital and diversify risk (Levine and Zervos 1998). Therefore, we expect that a more positive public perception of capital markets influences firms' financing strategies and consequently increases equity market size. We use this ratio under the assumption that equity market size is positively correlated with the ability to mobilize capital and diversify risk (Levine and Zervos 1998).

**Hypothesis 1.** *There is a positive relationship between public perception of capital markets and equity market size.*

### 3.3. Informal Institutions and Interrelations with Formal Institutions

Although formal institutions are relatively stable once established, they are not developed in isolation and are influenced by informal institutions such as shared cognitive understandings and acceptance of organizational practices in the national society (Muthuri and Gilbert 2011; Zucker 1987). Informal institutions influence the development of formal institutions by determining what problems are identified, their perceived importance, the generation of potential solutions for them, the evaluation of such solutions, and the behaviors enacted to implement those solutions (Zilber 2006; Park 2021). As formal institutions also reinforce a country's informal institutions (Inglehart and Baker 2000; Lewellyn and Bao 2014), these institutional factors coexist and cooperate (Pejovich 1999). Humphreys (2010) argues that market development must overcome the hurdle of regulatory legitimacy before contending the arena of public perceptions that adjudicate issues of normative and cognitive legitimacy. As a key component in the legitimate order, regulative legitimacy is most important during the first stages of legitimation (Scott 2008). La Porta et al. (1997) further found evidence that formal institutions are less effective in countries exhibiting low levels of trust among citizens. While formal institutional changes usually aim to bring new or reformed formal institutions closer to informal institutions, exogenously inspired reforms often create gaps between formal and informal institutions (Hartwell and Malinowska 2019). Hence, institutional reforms are more effective if informal institutions are developed in accordance with formal institutions. We propose that public perception of capital markets becomes more important if economic freedom already provides a certain level of stability and certainty to economic actors. Likewise, the more positive public perception of capital markets, the more pronounced is the effect of additional legal and regulatory improvements, as the overall stability and certainty for economic actors' increases. Therefore, we expect public perception of capital markets and economic freedom to interact in the explanation of equity market size, by mutually reinforcing each other.

**Hypothesis 2.** *Economic freedom positively moderates the relationship between public perception of capital markets and equity market size.*

### 3.4. Informal Institutions in Emerging Economies

Cross-national differences in institutional factors (see Figure 1), such as labor practices, banking systems, and subcontracting relationships, are key discriminating factors in the emergence and success of companies (Khanna and Palepu 1997; Williamson 1991). Through the change from a centrally planned to a market economy, transition economies have historically suffered from increased uncertainty for economic actors (North 1990). In transition economies, institutions are a prime determinant of the size of transaction costs, as firms are especially susceptible to environmental influences and changes (Tanas and Audretsch 2011). In weak institutional environments, informal institutions may substitute formal institutional mechanisms (Hartwell and Malinowska 2019). Knack and Keefer (1997) conclude that the effect of social trust on economic growth is higher in emerging countries in which informal institutions often substitute for or complement those established legally. As a result of EU accession, Poland is gradually importing formal institutions, but they remain relatively less significant compared to informal institutions, changing culture and societal perceptions (Maksimov et al. 2017). As capital markets work through expectations, the effect of institutions on capital market development may also depend on the trend of the institutional development (Berggren et al. 2012). As we expect that an increased level of uncertainty and a positive trend of institutional development is particularly present in a transition and emerging context, we propose that emerging and developed economies differ in the extent to which legitimacy influences capital market development. The Polish capital market, as a transition market, is expected to react more sensitively to changes in public perception of capital markets than the Austrian capital market.

**Hypothesis 3.** *The positive relationship between public perception of capital markets and equity market size is higher for Poland as a transition economy than for Austria as a developed economy.*

## 4. Data and Method

### 4.1. Sample and Estimation Methodology

To test our hypotheses, we use observations of the Austrian and Polish equity markets between Q1/2004 and Q1/2013. Due to the self-elaborated and time-consuming coding of newspaper articles, we limit the empirical research period. This period is especially appropriate because it covers the time when financial globalization became particularly salient (Doidge et al. 2013), and data availability and comparability were improved with the EU accession of Poland. Our data are compiled from several sources: equity market and macroeconomic data are taken from the Eurostat database, economic freedom is measured by the Heritage Foundation's Index of Economic Freedom, and public perception of capital markets is measured by using content analysis of newspaper articles from the Factiva database. We use newspaper articles because media publications are a good source of the general public's evaluation of the desirability and normativity of a concept. More than merely raising audiences' attention and increasing the salience of an issue on the public's agenda, the media record social knowledge and reflect and influence the values and concerns of a community (Hilgartner and Bosk 1988; McCombs and Shaw 1972; Meyer and Höllerer 2016). The media are not only a mirror of reality, but also the "site on which various social groups, institutions, and ideologies struggle over the definition and construction of social reality" (Gurevitch and Levy 1985, p. 19). The measurement of public perception of capital markets suggests that the public sphere is the primary space where meaning is constructed and negotiated, problems and solutions are discussed, and responsibilities and competences are contested (Meyer and Höllerer 2010). Although the content of newspaper articles is constructed by the interests and agendas of particular journalists, in the aggregate, it can be and has been used as a reliable indicator of generalized public perceptions (Deephouse 1996). Unlike niche communications, such as blogs, magazines, or legal documents, this kind of mass-media discourse has particular relevance when studying legitimacy because it both reflects and influences public perceptions (Humphreys 2010).

Our dataset includes 74 observations for Austria and Poland, where all variables are measured at the country level.[1] To account for within and between variations among observations, we use a fixed effects linear regression model (the Hausman test confirms that the difference in coefficients are not systematic, and a fixed effects model should be preferred). Because of the panel nature of the data, intertemporal correlation between the error terms may create contemporaneous correlation, which violates an important assumption of ordinary least squares regression (Beck and Katz 1995). Therefore, we run a Prais–Winsten regression with panel-corrected standard errors, which allows us to control for first-order auto-correlation.

As the period studied comprises a turbulent phase for capital markets, especially because of the financial crisis 2008–2009, we also account for time-variant omitted variables by including time fixed effects in our analysis. Furthermore, to capture a possible reverse causality problem in the regression analysis (Glaeser et al. 2004), we use content coding (see coding instructions in the Appendix A) and a lagged dependent variable, which will mitigate at least some of the endogeneity (Berggren et al. 2012). We measure capital market development at time *t,* and we use our independent variables at time $t - 1$. For interpretation purposes, we mean-center public perception of capital markets and economic freedom.

### 4.2. Variable Measurement

*Equity market size.* We compute equity market size using the quarterly value of listed shares divided by real GDP per year (base year 2005). We use this ratio under the assumption that equity market size is positively correlated with the ability to mobilize capital and diversify risk (Levine and Zervos 1998).



*Public perception of capital markets.* To measure public perception of capital markets, we evaluated newspaper articles from the Factiva database containing the keyword *"capital market"* (or abbreviations) in their title or lead paragraph (*n* = 3244). We chose multiple newspapers for corroboration (Golder 2000). We identified five leading newspapers, *"Der Standard"*, *"Die Presse"*, and *"Das Wirtschaftsblatt"* in Austria and *"Rzeczpospolita"* and *"Gazeta Wyborcza"* in Poland, which were available from the Factiva database throughout the period of investigation. We employed two categorical variables: "content" and "sentiment". Using Weber's (1990) coding protocols, the category "content" was used to identify articles that capture public perception of capital markets. We eliminated articles acquired through a data-extraction failure or a misleading keyword. Additional articles were eliminated that displayed not the public perception of capital markets but rather the ex post coverage of capital market development (capital market reports and corporate facts). This "content" coding reduced the final sample to 1556 articles. Of these articles, 978 were related to the Polish capital market and 578 to the Austrian capital market. Subsequently, we coded each newspaper article according to the sentiment it conveyed. We argue that although the evaluative standpoint is often a consequence of interpretive packaging, in principle, each article can be used to construct pro, contra, or neutral interpretations (Meyer and Höllerer 2010). Thus, we coded each article according to the position it adopted on capital markets (favorable, unfavorable, or ambivalent/neutral). To control for the reliability of the coding, we measured inter-coder reliability using Cohen's kappa values, where all values are at a substantial level (Landis and Koch 1977). Based on the sentiment coding of the identified articles, we calculated Janis–Fadner coefficients of imbalance (Deephouse 1996):

$$\text{Janis-Fadner coefficient} = (e^2 - ec)/(e + c)^2 \text{ if } e > c; (ec - c^2)/(e + c)^2 \text{ if } e < c; 0 \text{ if } e = c,$$

where e is the annual number of favorable articles and c is the annual number of unfavorable articles.

The Janis–Fadner coefficient ranges from $-1$ to $+1$, where a high presence of favorable articles yields a value closer to $+1$, and a high presence of unfavorable articles yields a value closer to $-1$. To consider the rather stable development of cognitive structures in national society and a possible bias through periods with few articles, we calculate yearly Janis–Fadner coefficients (for example, the Janis–Fadner coefficient for Q4/2003 is calculated by taking the coded articles between Q1/2003 and Q4/2003, and the subsequent periods accordingly).

*Economic freedom.* Economic actors generally inform their decisions by evaluating the overall formal institutional environment rather than evaluating a single aspect only (Slangen and van Tulder 2009). Formal institutions conceptualized as economic freedom can be defined by a holistic set of attributes of legislation, regulation, and legal systems that condition freedom of transacting, security of property rights, and transparency of government and legal processes (Globerman and Shapiro 2003). For example, if corruption is high and governments are unable to implement policies that enforce contractual rights, the level of uncertainty will be higher, requiring issuers to bear more risk associated with the related transactions (Chadee and Roxas 2013). If the level of economic freedom is high, firms can be sure that the valuation of the firm is fair and legitimate (Engelen and van Essen 2010). Thus, greater economic freedom encourages capital market development by reducing uncertainties for issuing firms, resulting in less potential for opportunism, lower transaction costs, and greater possibilities for achieving the desired rewards (North 1990). The use of an aggregated measure also allows us to capture the multifaceted nature of formal institutions. We use the Heritage Foundation's Index of Economic Freedom, which is based on 10 quantitative and qualitative factors, grouped into four broad categories, or pillars, of economic freedom. These are "Rule of Law" (property rights, freedom from corruption), "Limited Government" (fiscal freedom, government spending), "Regulatory Efficiency" (business freedom, labor freedom, monetary freedom), and "Open Markets" (trade freedom, investment freedom, financial freedom). A country's overall score is derived by averaging the 10 economic freedoms, graded each on a scale of 0 to 100, with

equal weight given to each. The Economic Freedom Index is measured on a yearly basis (assumed to be constant for the individual quarters of the year), again reflecting the rather stable concept of institutional change (for example, the Heritage Foundation's Index of Economic Freedom for the year 2004 is used for Q1/2004, Q2/2004, Q3/2004 and Q4/2004, and so on).

*Control variables.* We include several time-varying macroeconomic factors found to be relevant for capital market development. Previous research has demonstrated that *income* has a positive impact on capital market development, as higher income is usually associated with more flourishing listed firms that increase the propensity to invest in the equity market (Billmeier and Massa 2009; Smaoui et al. 2017). Income is measured as real GDP per capita per quarter (in constant Euros thousandth, base year 2005). Furthermore, *inflation* is commonly included as a proxy of macroeconomic stability. Inflation is measured as the change in consumer price index per quarter (base year 2005) in percent. In particular, moderate to high inflation may discourage financial intermediation and encourage saving in real assets (Chinn and Ito 2006). Finally, we control for country- and time-fixed effects to account for country-specific and several transnational influences on capital market development by including a country dummy (0 for Austria and 1 for Poland) and *time dummies* (per quarter). Table 1 provides a detailed overview of the regression variables.

**Table 1.** Variable definitions.

| Variable | Definition |
| --- | --- |
| Equity market size | Equity market capitalization as a share of real GDP per year (base year 2005) |
| Public perception of capital markets | Janis–Fadner coefficient of coded newspaper articles from Factiva database |
| Economic freedom | Heritage Foundation's Index of Economic Freedom 2014 edition |
| Income | Real GDP per capita in constant EUR thousandth (base year 2005) |
| Inflation | Change in consumer price index (base year 2005) in percent |
| Country dummy | Country dummy with Austria = 0; Poland = 1 |

## 5. Results

### 5.1. Descriptive Statistics

Tables 2 and 3 report descriptive statistics and correlations for the regression variables. On average, equity market size is similar between Austria and Poland. Public perception of capital markets as a proxy for informal institutions is higher in Poland than in Austria. Even during the financial crisis, Poland shows a positive public perception of capital markets, which constantly exceeds that in Austria. As expected, economic freedom is higher in Austria, with its moderately good to good formal institutions, than for Poland, with its mostly weak to moderately good formal institutions throughout the period of investigation.[2] The additional control variables show values that are basically in line with our expectations when comparing a developed with a transition economy, as prior research indicated that developed economies experience higher income and lower inflation rates (Billmeier and Massa 2009; Chinn and Ito 2006).

**Table 2.** Descriptive statistics.

| | Full Sample (n = 74) | | | | Austria (n = 37) | | | | Poland (n = 37) | | | | *t*-Tests |
|---|---|---|---|---|---|---|---|---|---|---|---|---|---|
| | Mean | S.D. | Min | Max | Mean | S.D. | Min | Max | Mean | S.D. | Min | Max | |
| Equity market size | 32.86 | 10.95 | 16.27 | 60.54 | 34.30 | 12.82 | 19.36 | 60.54 | 31.41 | 8.62 | 16.27 | 50.46 | 1.24 *** |
| Public perception of capital markets | 0.30 | 0.27 | −0.34 | 0.79 | 0.16 | 0.26 | −0.34 | 0.69 | 0.44 | 0.19 | 0.06 | 0.79 | −5.19 *** |
| Economic freedom | 65.71 | 5.20 | 58.10 | 71.90 | 70.53 | 1.47 | 67.60 | 71.90 | 60.89 | 2.22 | 58.10 | 64.20 | 22.03 *** |
| Income | 4.80 | 2.99 | 1.50 | 8.10 | 7.76 | 0.28 | 7.20 | 8.10 | 1.84 | 0.20 | 1.50 | 2.10 | 103.89 *** |
| Inflation | 0.66 | 0.64 | −0.64 | 2.49 | 0.55 | 0.50 | −0.45 | 1.97 | 0.77 | 0.74 | −0.34 | 2.49 | −1.46 |

Significance levels (two-tailed): * $p < 0.10$; ** $p < 0.05$; *** $p < 0.01$. Time dummies are not included for the sake of parsimony (for variable definitions see Table 1).

**Table 3.** Correlations.

| | 1 | | 2 | | 3 | | 4 | 5 |
|---|---|---|---|---|---|---|---|---|
| 1 Equity market size | 1 | | | | | | | |
| 2 Public perception of capital markets | 0.4775 | *** | 1 | | | | | |
| 3 Economic freedom | 0.1560 | | −0.5219 | *** | 1 | | | |
| 4 Income | 0.1422 | | −0.5340 | *** | 0.9518 | *** | 1 | |
| 5 Inflation | −0.0394 | | 0.1133 | | −0.1497 | | −0.1602 | 1 |

Significance levels (two-tailed): * $p < 0.10$; ** $p < 0.05$; *** $p < 0.01$. Time dummies and the country dummy are not included for the sake of parsimony (for variable definitions, see Table 1).

*t*-tests (two-tailed) show that, on average, public perceptions of capital markets, economic freedom and income are significantly different between Austria and Poland. We do not find statistically significant differences for equity market size and inflation. Hence, comparing the differences between Austria and Poland suggests that the development of equity market size cannot be fully explained by prior literature indicating the level of formal institutions measured by economic freedom and other macroeconomic factors, which are all in support of the Austrian capital market. As hypothesized, public perception of capital markets may be the missing piece of the puzzle in explaining the development of equity market size in Austria and Poland.

Studying the variables' correlations in Table 3, we find a significant correlation between equity market size and public perception of capital markets already supporting Hypothesis 1 without control variables. Moreover, income is highly correlated with economic freedom, confirming prior research arguing that income is usually associated with better formal institutions (Billmeier and Massa 2009).

*5.2. Regression Results*

To test our hypotheses, we conduct a hierarchical regression analysis with panel corrected standard errors. The models are calculated as follows:

(1a)　$\text{EMS}_{it} = \beta_0 + \beta_1\,\text{PPCM}_{it} + \beta_2\,\text{EF}_{it} + \beta_3\,C + \beta_{4+(t-1)}\,T_{t-1} + \varepsilon_{it}$,

(1b)　$\text{EMS}_{it} = \beta_0 + \beta_1\,\text{PPCM}_{it} + \beta_2\,\text{EF}_{it} + \beta_3\,\text{INC}_{it} + \beta_4\,\text{INF}_{it} + \beta_5\,C + \beta_{6+(t-1)}\,T_{t-1} + \varepsilon_{it}$,

(2)　　$\text{EMS}_{it} = \beta_0 + \beta_1\,\text{PPCM}_{it} + \beta_2\,\text{EF}_{it} + \beta_3\,(\text{PPCM}_{it}{*}\text{EF}_{it}) + \beta_4\,C + \beta_{5+(t-1)}\,T_{t-1} + \varepsilon_{it}$,

(3)　　$\text{EMS}_{it} = \beta_0 + \beta_1\,\text{PPCM}_{it} + \beta_2\,\text{EF}_{it} + \beta_3\,(\text{PPCM}_{it}{*}C) + \beta_4\,C + \beta_{5+(t-1)}\,T_{t-1} + \varepsilon_{it}$,

where $\text{EMS}_{it}$ is equity market size, $\text{PPCM}_{it}$ is public perception of capital markets, $\text{EF}_{it}$ is economic freedom, $\text{INC}_{it}$ is income, $\text{INF}_{it}$ is inflation, $C$ is the country-fixed effect and $T_{t-1}$ are time-fixed effects.

Table 4 presents the hierarchical regression results: Model 1a contains the institutional research variables only, Model 1b further adds the control variables, Model 2 adds the interaction effect of informal with formal institutions, and Model 3 adds the interaction effect of informal institutions with country. According to the presence of multicollinearity, Model 2 and Model 3 are calculated separately, without control variables, and in comparison to Model 1a.

**Table 4.** Hierarchical regression results.

| | Equity Market Size | | | | | | | |
| --- | --- | --- | --- | --- | --- | --- | --- | --- |
| | Model 1a | | Model 1b | | Model 2 | | Model 3 | |
| *Independent variables* | | | | | | | | |
| H1: Public perception of capital markets | 4.7958 | ** | 5.4003 | ** | 5.2572 | ** | 12.3374 | *** |
| Economic freedom | 1.5171 | *** | 1.5911 | *** | 1.1307 | *** | 1.4272 | *** |
| *2-way interaction terms* | | | | | | | | |
| H2: Economic freedom x Public perception of capital markets | | | | | 1.8491 | *** | | |
| H3: Country dummy x Public perception of capital markets | | | | | | | −14.2853 | +++ |
| *Control variables* | | | | | | | | |
| Income | | | −0.1489 | | | | | |
| Inflation | | | −0.1536 | | | | | |
| Country dummy | 10.2819 | *** | 9.9745 | | 6.3877 | * | 9.1921 | ** |
| Time dummies | YES | | YES | | YES | | YES | |
| Wald-Chi$^2$ | 554.80 | *** | 565.19 | *** | 812.70 | *** | 693.25 | *** |
| R$^2$ | 0.8701 | | 0.8722 | | 0.9080 | | 0.8938 | |
| Change in R$^2$ | | | 0.0021 | | 0.0379 | | 0.0237 | |
| Observations (n) | 74 | | 74 | | 74 | | 74 | |

Significance levels (two-tailed): * $p < 0.10$; ** $p < 0.05$; *** $p < 0.01$; +++. $p < 0.01$ in opposite direction than anticipated. Country-fixed and time-fixed effects regression analysis with panel-corrected standard errors (first-order autocorrelation and panel-level heteroskedastic errors). Time dummies are not included for the sake of parsimony (for variable definitions see Table 1). To assess whether multicollinearity poses a threat to the validity of our results, we computed variance inflation factors (VIF). VIF coefficients are all well below 4.0 for the simple model and the added two-way interaction. An exception is a larger variance inflation factor (VIF) for Economic freedom and the Country dummy. These VIFs should be interpreted with caution, as in our data there are a small number of cases in each category (dummy variables) that potentially have high VIFs, regardless of whether the categorical variables are correlated with other variables. Hence, multicollinearity does not necessarily exist. The results remain qualitatively unchanged when we re-run the model without Economic freedom as an additional control variable. These results suggest that multicollinearity is of no concern to their validity.

Model 1a and Model 1b in Table 4 provide empirical support for Hypothesis 1 and indicate a positive relationship between public perception of capital markets and equity market size (β = 4.7958, $p < 0.05$; β = 5.4003, $p < 0.05$). Therefore, public perception of capital markets as a proxy for informal institutions and equity marked size are indeed significantly and positively related. Model 2 and Model 3 add the proposed interaction effects. Hypothesis 2 suggests that the relationship between public perception of capital markets and equity market size depends on the level of economic freedom. The results confirm Hypothesis 2 and display a positive and significant interaction effect (β = 1.8491, $p < 0.01$). Finally, Model 3 contradicts Hypothesis 3 and shows that the positive relationship between public perception of capital markets and equity market size is more pronounced for Austria than for Poland (β = −14.2853, $p < 0.01$). This can be interpreted as indicating that the low level of public perception of capital markets in Austria prevent the Austrian equity market to grow in the same way as the Polish equity market. In this sense, informal institutions such as public perception of capital markets are important preconditions for capital market development. Compared to Model 1a, inclusion of the interaction effects increases the explained variance (ΔR$^2$ = 0.0379; ΔR$^2$ = 0.0237)[3].

### 5.3. Robustness Checks

We also run regressions by using a fixed effects model without panel-corrected standard errors and a feasible generalized least squares estimation. Furthermore, we use *equity market liquidity*, measured by the total value of trades divided by GDP, as an additional measure of capital market development, which is generally associated with market efficiency (Levine and Zervos 1998). We also run regressions by including further and removing

existing control variables. We calculate regressions replacing economic freedom with trade openness (Rajan and Zingales 2003), measured by the ratio of quarterly real exports plus real imports to real GDP (base year 2005). We additionally use the percentage of leftwing members of parliament as a measure of *political leadership* and the percentage of households with internet access as a measure of internet infrastructure (both are measured on a yearly basis, assumed to be constant for the individual quarters of the year) and exclude income and the country dummy in light of high multicollinearity. We are further aware of using mixed-frequency data within our quarterly analyses, as the Heritage Foundation's Index of Economic Freedom is only available on a yearly basis. Although we are convinced that the stable development of the index mitigates a possible measurement error and related problems in the regression analyses due to the chosen distribution scheme of the Economic freedom variable, we further run the regression analyses by assuming a uniform quarterly adjustment of the yearly index. The results in all robustness checks show qualitatively comparable results corresponding to our main findings.

## 6. Discussion and Implications

### 6.1. Discussion and Research Implications

We identify public perception of capital markets as a novel and dynamic measure for the workings of informal institutions within national societies, one that influences equity market size within and across countries. Building on arguments that informal institutions play a pivotal role in how individuals and firms engage in, or disengage from, economic transactions (North 1990), we find evidence that informal institutions affect equity market size beyond the influence of formal institutions. Hence, we contend that public perception of capital markets plays an important role in motivating and constraining firms' financing strategies.

Our study is also closely related to existing research that has shown the effect of individual formal institutional factors, such as minority shareholder protection, on capital market development (Djankov et al. 2008; La Porta et al. 2000). Our results confirm that economic freedom reduces uncertainties for issuing firms and consequently increases equity market size. Because the importance of formal institutions to aggregated economic outcomes has already been discussed in prior research (Billmeier and Massa 2009; Holmes et al. 2013), we focus on examining the concurrent roles of formal and informal institutions in explaining within- and between-country variations in capital market development. By including public perception of capital markets and economic freedom, we add to the understanding of the complex interplay between formal and informal institutions. Finding that both institutional factors are mutually reinforcing each other, we add further insights to the inconclusive empirical findings on the impact of institution on economic outcomes (Cruz-García and Peiró-Palomino 2019; Holmes et al. 2013; Lewellyn and Bao 2014). Where regulatory and legal stability for economic actors is high, public perception of capital markets plays a larger role in conferring market legitimation. Our results reveal that formal and informal institutions act in concert and are most effective when developed jointly. However, we admit that the results depend on an empirical setting with certain formal institutions in place. Therefore, our findings are in line with the perspective that different institutional factors interact in complex and unique ways to enable and guide patterns of behavior that yield variation in economic outcomes.

We also contribute to the academic debate on the relationship between informal institutions and capital market development, with the idea that public perception of capital markets matters differently within developed and emerging economies. Reducing the uncertainty for firms through an increased public perception of capital markets is more important for Austria, with its low level of public perception of capital markets in place.

### 6.2. Policy Implications

Our study reveals that managers interact with the institutional environment in discovering, evaluating, and exploiting opportunities. The integration of the global capital market has enabled firms to more easily access foreign capital markets by listing on foreign stock

exchanges (Bell et al. 2012; Tupper et al. 2018). Cross-listing widens the pool of potential investors, helps to overcome home-market institutional weaknesses for raising capital, and may help to extend firms geographic scope (Lindorfer et al. 2016). Understanding how to utilize the institutional environment within different capital markets can be critical to firms' global efficiency and competitiveness.

Our arguments and findings also have meaningful implications for policy makers in helping to clarify the circumstances of capital market development, whereby formal and informal institutions cooperate to support firms' availability to external finance. In order to establish an attractive capital market, the political sphere needs to establish an institutional environment where formal and informal support coexists. Consequently, establishing an internationally competitive capital market requires a supportive society in addition to regulatory reforms. Our interview study confirms this finding. In particular, the close relationship and collaboration between the Polish government (e.g., Ministry of Treasury) and the Warsaw Stock Exchange seem to have been a key driver of the Polish success story in recent years:

> "If we try to cover the privatization transactions without the politicians' support, it's going to be impossible... our minister of the state, as a main shareholder of the Warsaw Stock Exchange, is giving us support."

### 6.3. Limitations and Future Research

Our study builds on the self-elaborated and time-consuming coding of 3244 newspaper articles, limiting the empirical sample to two countries. In that respect, we acknowledge the trade-off between a more superficial analysis using a large dataset with many countries and a deeper analysis using a limited number of countries. Furthermore, we consider Austria to be a developed economy and Poland to be a transition economy. Although we believe that our inferences are transferable to a wider geographic scope, our results should be interpreted with caution. Poland accords with capitalist economies in many of its institutional aspects and relies on the use of markets for economic coordination (Redek and Sušjan 2005). Future research may examine whether a certain formal institutional threshold is a prerequisite to enabling the influence of informal institutions on capital market development. Moreover, we fully rely on macroeconomic data, not taking individual firm perceptions into account. We believe that research providing insights into the perceptions associated with listing decisions would greatly augment our current understanding of the links between cognitive structures and economic behavior. The measurement of public perception of capital markets entails a possible reverse-causality issue, where equity market size may also influence public perception of capital markets. We are fully aware of this endogeneity problem and approach it within the coding process and within the data analysis as accurately as possible. Furthermore, although the inter-coder reliability levels are sufficient, a possible bias due to personal coding may still exist. Hence, future research may bypass personal coding and explore how other informal measures may influence capital market development as well as how they may interact with formal institutions in explaining important financial phenomena. Similarly, we conceptualize formal institutions as economic freedom. We believe this proxy is particularly appropriate for our study, since it is more likely to capture the significant effects imparted to issuing firms (Lewellyn and Bao 2014). However, future research could also disaggregate formal institutional measures.

**Author Contributions:** Conceptualization, R.L., A.d. and I.F.; methodology, R.L. and I.F.; software, R.L.; validation, R.L.; formal analysis, R.L. and I.F.; investigation, R.L. and I.F.; resources, A.d. and I.F.; data curation, R.L.; writing—original draft preparation, R.L. and I.F.; writing—review and editing, R.L. and A.d.; visualization, R.L.; supervision, A.d. and I.F., project administration, R.L., A.d. and I.F.; funding acquisition, A.d. and I.F. All authors have read and agreed to the published version of the manuscript.

**Funding:** This work was supported by the Jubiläumsfonds der Stadt Wien für die Wirtschaftsuniversität Wien (there is no grant number applicable).

**Institutional Review Board Statement:** Not applicable.

**Informed Consent Statement:** Not applicable.

**Data Availability Statement:** Our data are compiled from several sources: equity market and macroeconomic data are taken from the Eurostat database (https://ec.europa.eu/eurostat/data/database, accessed on 10 December 2022), economic freedom is measured by the Heritage Foundation's Index of Economic Freedom (https://www.heritage.org/index/, accessed on 10 December 2022), and public perception of capital markets is measured by using content analysis of newspaper articles from the Factiva database (https://www.dowjones.com/professional/factiva/, accessed on 10 December 2022). The latter are available on request from the corresponding author.

**Acknowledgments:** We would like to thank the editor and two anonymous reviewers, the organizers and participants of the 1st Contemporary Issues in Emerging Markets Conference (CIEMC) 2022, Stefan Edlinger-Bach (Vienna University of Economics and Business), and Philipp Hampl (Vienna University of Economics and Business) for their comments and feedback that have improved the quality of the paper.

**Conflicts of Interest:** The authors declare no conflict of interest.

## Appendix A. Content Coding

**Table A1.** Public perception of capital markets: Content coding.

| *Public perception of capital markets: Content coding.* |
| --- |
| The aim of categorizing the newspaper articles according to the conveyed content lies in the complex task of considering relevant articles only. Thus, content coding may prevent using articles that may bias the resulting variable measurement. Because the aim is to capture *public perception of capital markets*, the analysis should only consider articles that reflect the public discourse and are not displaying ex post information on capital markets or capital market actors. Therefore, the following categories and respective descriptions are provided to select those articles that are not reflecting *public perception of capital markets*: |
| **"Data extraction failure"** comprises articles that should not have been displayed by the Factiva database. This category comprises articles that are textually precisely the same as another article already considered for the analysis. |
| **"Misleading keyword"** comprises articles that are not related to capital markets, although the keyword is mentioned in the title or lead paragraph. Examples within this category can be manifold and comprise, for example, articles about (1) country ratings or (2) bank bailouts. |
| **"Capital market report"** comprises articles that display capital market related ex post information. These articles are characterized by a tabular or abbreviated form and comprise facts on capital market activities such as (1) stock tickers, (2) bond tickers, and (3) index developments. |
| **"Corporate fact"** comprises articles that display firm related ex post information. These articles comprise facts on corporate activities such as (1) capital market performance, (2) share issues, and (3) capital increases. |

## Notes

[1] As both countries of analysis only have one stock exchange, we do not have to differentiate between country- and exchange-level data.

[2] The Heritage Foundation considers countries that score in the 80–100 range as having the best ("most free") institutions; those in the 70–79.9 range as having good institutions, and those in the 60–69.9 range as having moderately good institutions. Countries in the 50–59.9 range are characterized by mostly weak institutions, and countries in the 0–49.9 range have the weakest institutions.

[3] For Model 2 and Model 3 results of an F-test further indicate that the coefficients of our interaction variables are significantly different from zero ($p < 0.001$ for Model 2; $p < 0.001$ for Model 3).

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
