# Peer review of "When Institutional Plates Collide: The Dynamic Impact of Informal Institutions on Capital Market Development"

_jrfm, doi:10.3390/jrfm16030178_

Round 1
Reviewer 1 Report
REVIEW OF SUBMISSION TO JOURNAL OF RISK AND FINANCIAL MANAGEMENT – 2127116 - “WHEN INSTITUTIONAL PLATES COLLIDE: THE DYNAMIC IMPACT OF INFORMAL INSTITUTIONS ON CAPITAL MARKETS DEVELOPMENT”
Summary of the paper
This paper uses the lens of institutional theory to investigate the role of informal institutions, in facilitating capital market development.
The paper uses data from Austria and Poland, motivated by several institutional differences between these two countries. Austria has a regulatory régime based on European Code Law; Poland is adopting traditional features of Anglo-American common law regulatory regimes. Austrian listed companies tend to be subject to higher shareholdings by insiders than their Polish counterparts. Austria has a mature, stagnant capital market; conversely, the Polish capital market is growing, in the aftermath of the fall of Communism. These differences indicate that using data from these two countries would be suitable for investigating the research question.
Three hypotheses are tested. The first hypothesis postulates a positive association between public perception of capital markets and capital market size. The second hypothesis conjectures that this positive association is positively moderated by economic freedom. The third hypothesis postulates that the positive association is stronger in Poland than Austria.
The methodology entails a variant of Ordinary Least Squares regressions, with standard errors corrected for time-series autocorrelation. Country-quarter observations are used, over the investigation period 2004Q1 to 2013Q1. The dependent variable, equity market size, is a proxy for equity market development. The independent variables of interest include a country dummy (assuming the value of 1 (0) for Polish (Austrian) observations), a measure of public perception of capital markets (based on the sentiment in newspaper articles) and the Heritage Foundation Index of Economic Freedom. The metric of public perception of capital markets is interacted with the measure of economic freedom (to test the second hypothesis) and the dummy flagging Polish observations (to test the third hypothesis). The controls include fixed year effects, income (real Gross Domestic Product), inflation (quarterly change in Consumer Price Index) and a measure of trade openness. The final sample comprises 74 country-quarter observations. Data sources include Eurostat and Factiva.
The authors interpret their results as supporting all three hypotheses. The findings are robust to a battery of sensitivity tests. The authors interpret the evidence as suggesting that informal institutions augment formal institutions, in facilitating the development of capital markets.
Critical review
Motivation and introduction
The authors motivate their paper by noting that the literature is replete with evidence about the role of formal institutions (such as national regulatory régime, use of common- versus code-law and government efficiency). This evidence needs to be complemented by investigation of the complementary role play be informal institutions (such as public confidence in the capital market) in facilitating capital market development. I commend the authors for convincingly arguing the paper’s motivation.
The structure and content of the introduction should be improved. The penultimate paragraph of the introduction summarises the study. This should receive more emphasis. The first two paragraphs of the conclusions, discussing unique features of the study, belong in the introduction. Similarly, a short summary of the findings should be in the introduction.
Literature review
A weakness of this paper is the lack of a separate literature review. The entirety of the introduction, preceding the penultimate paragraph, belongs in the literature review. Similarly, the section of the conclusions, discussing how the paper contributes to the literature on the role of informal institutional factors in facilitating capital market development, also belongs in the literature review.
Discussion of institutional features of Austria and Poland
The case for using data from these two countries is argued convincingly.
I can suggest two more references that would be appropriate, for discussion of the Polish institutional environment. Grosfeld and Hashi (2007) explain that there is still an element of the development state ideology, underpinning the post-Communist regulatory régime in Poland. Dobija and Klimczak (2010) provide examples of typical Anglo-American features that have been embraced by Polish regulators. The latter study also provides evidence that the quality of earnings, produced by Polish companies, has not improved over different regulatory sub-periods in the post-Communist era. A possible interpretation is that Polish companies had always produced high quality earnings since the earliest sub-period of the study, which began in 2001. This finding suggests that companies in Poland and Austria may have had comparable earnings quality, during your investigation period. It reduces concern that the results are due to unobserved differences between Poland and Austria, correlated with earnings quality.
Underpinning theory and hypothesis development
These dimensions of the paper are very sound.
I have two minor suggestions for improvement. In the section on variable measurement, the authors explain that equity market size is positively related to the capacity to mobilise capital and diversify investment risk. This explanation should be re-located to the discussion preceding H1. The discussion preceding H3 should cite Figure 1, presenting evidence that economic freedom is higher in Austria than Poland.
Methodology
The methodology is extremely sound, with the exception of three concerns.
The explanation in Endnote (2) belongs in the text. It clarifies that the measure of overall public sentiment is calculated on a country-quarter basis, using data from the previous year.
The decision to estimate panel-corrected standard errors, to correct for temporal correlation, is wholly appropriate. However, the discussion in the paragraph in Section 4.1, confounds heteroscedasticity and autocorrelation. These are two distinct issues. The authors should eliminate mention of heteroscedasticity.
I also suggest deleting the control for trade openness. This variable seems to be captured within the index of economic freedom.
Descriptive statistics
I have two concerns.
Firstly, Table 2, presenting univariate statistics, shows that equity market sizes are comparable, between Poland and Austria. This finding warrants more attention. The authors should report the t-statistic, of mean equality. They should discuss this statistic in the text. It further ameliorates concern that the results may be driven by other differences between Poland and Austria, correlated with equity market size.
Secondly, the authors should discuss the significant bivariate correlations in Table 3. Possible economic interpretations would be particularly important.
Empirical results
I have three concerns.
Firstly, throughout the text, the authors have overstated support for H1. In Table 4, the coefficient attaching to the proxy for public perception of capital markets is significant in only one model of two. This finding is echoed in the results of the sensitivity analyses, currently in the appendix.
Secondly, the discussion should mention that support for H3, in the sensitivity analyses, is slightly weaker than in the body of the paper.
Thirdly, Appendix B should be deleted. Discussion of the sensitivity results, within the body of the paper, would be sufficient.
Presentation
The presentational quality of some of the tables should be improved. The numerals should be right-aligned within a cell and occupy only one line of the cell.
Recommendation
This paper reflects a high quality. The concerns I have broached could be readily addressed. My most critical concern relates to the control for trade openness. I recommend that the authors be invited to re-submit a substantially revised version of the manuscript to Journal of Risk and Financial Management.
References, not cited in the paper
Dobija, D. and K. Klimczak, 2010, “Development of Accounting in Poland: Market Efficiency and the Value Relevance of Reported Earnings”, The International Journal of Accounting 45 (3), 356-374.
Grosfeld, I. And I. Hashi, 2007, “Changes in Ownership Concentration in Mass Privatised Firms: Evidence from Poland and the Czech Republic”, Corporate Governance – An International Review 15 (4), 520-534.
Author Response
Please, see the attachment.

Reviewer 2 Report
The paper discusses an interesting issue related to the impact of informal institutions on capital market development. According to the authors, the novelty of the study lies mainly in the dynamic approach to measuring informal institutions.
This article is an original research article; however, particularly the empirical sections (results and discussion) should be expanded and improved prior to publication.
Detailed remarks:
MAJOR:
1) The methodological part related to the econometric modelling requires improvement.
· First of all, provide the regression equation.
· The methodology requires further justification for fixed and random effects.
· It is important to present the VIF test on multicollinearity between independent variables.
All these aspects that are not found in the paper represent weaknesses of the research.
2) Table 3 presents the correlation matrix. However, the results were not analysed and discussed. Some independent variables, such as Income, Trade Openness, and Economic Freedom are highly correlated (correlation coefficient above 0.9). It suggests the problem of multicollinearity not addressed in the paper.
3) A broader interpretation and commentary on the research results is needed. This part (Results) is quite superficial. In particular, the regression results should be analysed with more attention with respect to the statistical significance of the parameters. For example:
· “The control variables are in line with our expectations: economic freedom is positively, income is positively, inflation is negatively, and trade openness is positively associated with equity market size”. (p. 11)
However, the parameters are statistically significant only for economic freedom and trade openness.
· Public perception of capital markets is statistically significant only in Model 2, in Model 3 it is not statistically significant. Try to explain these results.
4) Figure 2: It is not clear to the reader. Furthermore, the figures were not analysed and explained in the paper.
5) At the same time, I consider that the conclusions part of the work should be expanded. The conclusions at the end of the paper should be expanded showing the economic policy implications of the research results.
MINOR:
1. The main contribution and novelty of the research should be stressed more in the abstract. In addition, brief information on the research period and all research methods used should also be added.
2. The main goal of the study is not clearly presented in the Introduction. It requires improvement. Moreover, I recommend to supplement the Introduction with a brief justification for the choice of the research period. This was done later in the study. However, from my point of view, giving the justification in the Introduction would make it easier for readers to follow the paper. Later on, please explain why the analyses end on 2013 (almost 10 years ago), because it makes the research seem out of date (time consuming content analysis?).
3. Figure 1: The figures would be clearer if the measurement units of individual variables were given. The sources of data should be listed below the figure.
Reviewer 3 Report
In the manuscript the authors empirically analyze how changing perception of capital market legitimacy influences capital market development in Austria and Poland. I have to admit that for me the title of the manuscript was a bit confusing: I was not sure on which aspects of informal institutions the authors plan to focus. By the very end of the Introductory section, I was still not absolutely sure whether the focus will be on trust, legitimacy, or something else. I know of literature dealing with legitimacy, i.e. perceptions of some institutions by public, and I suggest the authors use the word legitimacy right in the title to set the expectations right.
I am aware of the fact that available survey data capturing trust or legitimacy are of much lower frequency, but I still wondered if it would not be possible to provide some information analogical to "public perception of capital markets" from other sources. At least to see whether the cross-country differences obtained from the newspaper articles are present also in other data sources. It could strenghten the credibility of the estimated perception/legitimacy index.
I was wondering if the argument that "a more positive public perception of capital markets further increases the likelihood that other firms will behave similarly" (p. 5) is not, in fact, similar to the arguments presented by Deirdre McCloskey in her books on bourgeois virtues and their perception by the majority population. Maybe it could be used as another source of motivation of the authors' research question.
Regarding the used variables and methodology:
1. Looking at figure 1, it seems to me that income and economic freedom could be non-stationary. If they are, please elaborate if the use of non-stationary variables in your analysis does not constitute a problem. Maybe this is the reason why the income variable is insignificant?
2. It was not clear to me whether the content coding of newspaper articles has been done by hand or using some of the commonly-used tools for this task. After reading the whole manuscript I believe that it was done by hand, but I would appreciate a better and more detailed explanation.
3. I miss some measure of loan costs among the control variables, such as the main monetary policy interest rate. The reason is that bank loans can be perceived as an alternative source of corporate funding. And therefore have the potential to influence the equity market size.
4. I appreciate the discussion regarding potential reverse-causality or endogeneity problem in section 6.3. I noticed that you use lagged independent variables to tackle this issue. (Btw you write "lagged dependent variable" on p. 7.) First, I suggest that you explicitly write the regression specification as a mathematical expression to make this more visible. And second, wouldn't it be possible to at least provide a Granger causality test to see whether the public perception variable granger-causes equity market size? Even though I like the specifications of models you use (especially the interaction terms), it is uncertain whether you can really interpret the results causally.
Author Response
Dear Editor, we have received the notification of the major revision and the two reviews on January 15. At this day, I have downloaded the two review reports. The editor asked for a 10 days deadline which was extended upon request until today, February 14. I have never seen the third review report and we have never received a notification. The time stamp also documents that the review was uploaded later, January 16. We therefore ask for your understanding that we are not able to answer just in time to this late review report. In order to meet this deadline, we therefore do not provide specific comments on review 3. We are still convinced that our revision has addressed all major issues.

Round 2
Reviewer 1 Report
REVIEW OF SUBMISSION TO JOURNAL OF RISK AND FINANCIAL MANAGEMENT – 2127116 - “WHEN INSTITUTIONAL PLATES COLLIDE: THE DYNAMIC IMPACT OF INFORMAL INSTITUTIONS ON CAPITAL MARKETS DEVELOPMENT”
Preliminary clarifications
1. This report should be read in conjunction with my report from the previous round of this submission.
2. The essence of the paper has not fundamentally changed. Hence, I have not reproduced my summary of the paper.
Critical review
The authors have attended to almost all of the concerns I broached in my report from the previous round of submission. For each concern, the authors have either explained how they have revised the paper, to address the concern or lucidly argued why they should not address the concern.
I have two remaining criticisms, both relatively minor.
1. The results in Table 4 no longer support H3. However, the authors use the same symbol to denote the significance of Country dummy * Public perception of capital markets (in Model 3) as used to denoted the significance of Economic freedom * Public perception of capital markets (Model 2). This symbol is three asterixis, in superscript font. This is confusing since the coefficient of interest in Table 3 has the “wrong” polarity. To redress this concern, the authors should replace the three asterixis adjacent to Country dummy * Public perception of capital markets to three “plus” signs, also in superscript font (i.e., “+++”). They should then embellish the captions to the table, to explain that the latter symbol denotes significance at the one-percent level, in the opposite direction from anticipation.
2. There is some usage of colloquial expressions throughout the paper. Examples include “along with”, “come very close to”, “apart from”, “beyond that”, “gets more important”, “pick up”, “and so on”, “in line with” and “so-far”. The authors should replace these expressions with more formal counterparts.
Recommendation
I recommend that the paper be accepted for publication in Journal of Risk and Financial Management, subject to the authors attending to the aforementioned concerns, to the satisfaction of the editorial board.
Reviewer 2 Report
I would like to thank the authors for their time and preparation of a new version of the manuscript. I appreciate the authors' involvement in the preparation of the revised version of the paper.
I have analysed the paper very carefully and found that the authors have taken into account the recommendations made. As a result, the quality of the paper has improved. Hence, I recommend the acceptance of this article for publication.
Prior publication, however, I would suggest the following:
- In Table 5, estimation 1 differs from estimations 2 and 3 in including 2-way interaction variables (PPCM*EF and PPCM*C), and the authors provide the R2 for each estimation, but still a significant test of including these variables would be informative (Ftest on PPCM*EF and PPCM*C),
- Table 5: to increase the validity of estimations: a) the test on normal distribution of residuals would be needed; b) instead of showing mean VIF values, it would be more informative to provide minimum and maximum VIF values,
- verifying the Introduction and the theoretical part of the paper, because there are repetitions in these parts. For example:
p.7 “Following institutional theorists, we distinguish between formal and informal institutions: formal institutions refer to laws and regulations of a particular country, and informal institutions are supported by values, beliefs, customs, traditions, and codes of conduct (North, 1990;Salomon & Wu, 2012).”
pp. 9/10: “Institutional quality is supported by formal and informal institutional factors (North, 1990), where the former refers to laws, and regulations, and the latter to values, beliefs, customs, traditions, and codes of conduct (Salomon &Wu, 2012)”.
- Moreover, please check all abbreviations - they should be explained when used for the first time in the text (e.g. IFRS p.5).
Author Response
Please see the attachement.
